# FLS2–RBOHD–PIF4 Module Regulates Plant Response to Drought and Salt Stress

**DOI:** 10.3390/ijms23031080

**Published:** 2022-01-19

**Authors:** Zhixin Liu, Chenxi Guo, Rui Wu, Yunhe Hu, Yaping Zhou, Jiajing Wang, Xiaole Yu, Yixin Zhang, George Bawa, Xuwu Sun

**Affiliations:** 1State Key Laboratory of Crop Stress Adaptation and Improvement, School of Life Sciences, Henan University, 85 Minglun Street, Kaifeng 475001, China; zxlsch2019@163.com (Z.L.); chenxi1445@163.com (C.G.); wr_7538@163.com (R.W.); huyunhe2022@163.com (Y.H.); 15737165974@163.com (Y.Z.); wdj970731@163.com (J.W.); yxl86420@163.com (X.Y.); zyx99292021@163.com (Y.Z.); ge6229@gmail.com (G.B.); 2State Key Laboratory of Cotton Biology, School of Life Sciences, Henan University, 85 Minglun Street, Kaifeng 475001, China; 3Key Laboratory of Plant Stress Biology, School of Life Sciences, Henan University, 85 Minglun Street, Kaifeng 475001, China

**Keywords:** FLS2, RBOHD, PIF4, transcriptome, drought, salt, stress

## Abstract

As sessile organisms, plants are constantly challenged by several environmental stresses. Different kinds of stress often occur simultaneously, leading to the accumulation of reactive oxygen species (ROS) produced by respiratory burst oxidase homolog (RBOHD) and calcium fluctuation in cells. Extensive studies have revealed that flagellin sensitive 2 (FLS2) can sense the infection by pathogenic microorganisms and activate cellular immune response by regulating intracellular ROS and calcium signals, which can also be activated during plant response to abiotic stress. However, little is known about the roles of FLS2 and RBOHD in regulating abiotic stress. In this study, we found that although the *fls2* mutant showed tolerance, the double mutant *rbohd rbohf* displayed hypersensitivity to abiotic stress, similar to its performance in response to immune stress. An analysis of the transcriptome of the *fls2* mutant and *rbohd rbohf* double mutant revealed that phytochrome interacting factor 4 (PIF4) acted downstream of FLS2 and RBOHD to respond to the abiotic stress. Further analysis showed that both FLS2 and RBOHD regulated the response of plants to drought and salt stress by regulating the expression of *PIF4*. These findings revealed an FLS2–RBOHD–PIF4 module in regulating plant response to biotic and abiotic stresses.

## 1. Introduction

Unlike animals, plants are sessile and cannot avoid various adverse factors present in the surrounding environment by moving their position [1,2]. To survive and deal with several biotic and abiotic stresses, plants have evolved a set of sophisticated strategies [2,3]. Interestingly, numerous studies have revealed that plants can respond to different stresses by activating signaling pathways similar to those in animals, such as MAPK and calcium signals, or by inducing the synthesis of phytohormones or small molecular compounds [1,4,5]. This greatly enhances the vitality of plants to withstand environmental stresses. Under normal circumstances, different stress factors often occur simultaneously, increasing the extent of damage in plants. For example, drought can aggravate the damage caused by pests and increase the susceptibility of plants to infections, whereas excessive humidity leads to a more serious invasion by pathogenic bacteria [6,7,8,9,10,11,12]. Therefore, plants need to recognize or respond to biotic and abiotic stresses to deal with these environmental problems [13,14,15,16,17,18,19,20,21,22,23,24]. Recently, the functions of CLAVATA3/embryo-surrounding region-related (CLE) peptides and their receptors have been identified and characterized in regulating the pathogen invasion and drought stress of plants, suggesting that plants may utilize a single signaling pathway or strategy to respond to compound stress [25,26,27,28]. Several studies have reported that both biotic and abiotic stresses can lead to the accumulation of H_2_O_2_ and the activation of the Ca^2+^ signal pathway [29]. In particular, the synthetic signals of H_2_O_2_, such as salicylic acid, can diffuse from damaged leaves to healthy leaves, which, in turn, induce systematic defense signals in plants against local stress [30,31,32]. This phenomenon further suggests a similarity in signal transmission between biotic and abiotic stresses. Although with continuous breakthroughs in research, especially the identification and characterization of H_2_O_2_ and Ca^2+^ receptors [29,33,34,35], the molecular mechanism underlying how plants sense compound stresses has been deciphered, a gap still exists in understanding the mechanism underlying the perception and regulation of compound stress.

Flagellin sensitive 2 (FLS2) is one of the best-studied immune signal receptors [36,37] that activates the downstream immune signaling pathways and induces the production of reactive oxygen species (ROS) after recognizing microbe-associated molecular patterns (MAMP) [38,39,40,41,42,43,44,45]. Extracellular ROS is majorly generated through the evolutionarily conserved NADPH oxidase (NOX) family [46,47]. In plants, ROS can be produced by respiratory burst oxidase homolog (RbOH) protein [46]. *Arabidopsis thaliana* respiratory burst oxidase homolog D (RBOHD) functions as an essential regulator of ROS [48,49]. RBOHD dependent production of ROS is essential for activating the downstream signaling of immune response. RBOHD and receptor FLS2 are localized in the plasma membrane [50]. BOTRYTIS-INDUCED KINASE1 (BIK1), as a co-receptor of FLS2, can interact with RBOHD to regulate its phosphorylation and the immune response mediated by RBOHD [51]. During pathogen invasion, FLS2 interacts with Brassinosteroid Insensitive 1 (BRI1) associated kinase receptor 1 (BAK1) to form a functional complex, resulting in Ca^2+^ influx and ROS production and activation of the MITOGEN-ACTIVATED PROTEIN KINASE (MAPK) cascade pathway [45]. BAK1 can phosphorylate ubiquitin ligases PLANT U-BOX12 (PUB12) and PUB13, mediate the binding of FLS2–PUB12/13, and subsequently inhibit the activity of immune signaling by ubiquitination of FLS2 [40]. In addition, brassinosteroid signal kinase 1 regulates FLS2-mediated ROS burst through interaction with FLS2 [52]. Furthermore, jasmonate (JA) contributes to flg22-triggered immune response, as evidenced by a stronger flg22-triggered ROS burst and callose accumulation in JA signal mutants *jar1* and *coi1* than in the wild-type (WT) [53].

Several studies have proposed that Ca^2+^, JA, and ROS induced by these biotic stresses also regulate plants’ responses to abiotic stresses [14,15,17,19,20]. In addition, certain studies have found that FLS2- and RBOHD-mediated biotic stress signals can cause extensive changes in the transcriptome. However, little is known about the effects of immune and ROS signals on the transcriptome during abiotic stress. To investigate the potential effects of immune and ROS signals on the transcriptome during response to abiotic stress and reveal the possible mechanisms involving how this regulates abiotic response of higher plants, we analyzed the responses of *fls2* mutant and *rbohd rbohf* double mutant to drought and salt stress using RNA sequencing. We identified differentially expressed genes (DEGs) in the *fls2* mutant and *rbohd rbohf* double mutant and constructed corresponding transcription factor (TF) regulatory networks. These analyses identified a key function of PHYTOCHROME INTERACTING FACTOR 4 (PIF4) that acts downstream of FLS2. These findings revealed an FLS2–PIF4 module that regulates plant response to biotic and abiotic stresses. In addition, these results provide important experimental evidence for understanding the mechanism by which FLS2 and RBOHD regulate the response of plants to abiotic stress.

## 2. Results

### 2.1. Analysis of Sensitivity of fls2 and Rbohd Rbohf Mutants to Drought and Salt Stress

FLS2 and RBOHD play an important role in perceiving pathogen invasion [45,54,55]. To explore their potential roles in regulating plant response to abiotic stress, we investigated the sensitivity of *fls2* and *rbohd rbohf* mutants and WT to drought and salt stress. As shown in Figure 1A, no significant difference was observed between WT and mutants when grown on 1/2 MS medium plates. Under normal growth conditions, the growth of *rbohd rbohf* double mutant was slower than that of WT (Figure 1B). After 1-week drought treatment, the growth of *rbohd rbohf* double mutant was strongly inhibited (Figure 1B,C). Compared with WT, the leaves of *rbohd rbohf* double mutant exhibited yellowing following treatment with NaCl (Figure 1B). Surprisingly, the growth of *fls2* mutant showed less sensitivity to drought and salt stress (Figure 1A–C), suggesting that FLS2 may be involved in sensing drought and salt stress signals.

### 2.2. FLS2 and RBOHD Are Involved in the Regulation of H_2_O_2_ Production in Response to Drought and Salt Stress

Singlet oxygen produced by plants can induce chloroplast apoptosis and leaf yellowing. In addition, H_2_O_2_ can antagonize the singlet oxygen signal and resist chloroplast apoptosis and leaf yellowing induced by singlet oxygen under oxidative stress [56]. Considering the yellowing phenomenon of *rbohd rbohf* double mutant leaves under salt stress, we suggested that the *rbohd rbohf* double mutant could not produce H_2_O_2_, resulting in chloroplast destruction and degradation caused by the accumulation of cell singlet oxygen. We detected the levels of O^2−^, H_2_O_2_, and apoptosis in the leaves of *fls2* mutant, *rbohd rbohf* double mutant, and WT under different treatment conditions by 3, 3′-diaminobenzidine (DAB), nitro blue tetrazolium (NBT), and trypan blue staining. As shown in Appendix A, drought and salt stress treatments increased the accumulation of O^2−^ in WT. Compared with WT, the level of O^2−^ in the leaves of *rbohd rbohf* double mutant was lower under control, drought, and salt stress conditions, whereas the level of O^2−^ in the *fls2* mutant was similar to that in WT. As demonstrated in Appendix A, drought treatment in WT had little effect on H_2_O_2_ production; however, NaCl treatment significantly increased the accumulation of H_2_O_2_. After drought or NaCl treatment, the levels of H_2_O_2_ in both *fls2* mutant and *rbohd rbohf* double mutant were lower than that in WT. As shown in Appendix A, cell death was increased in the leaves of WT, and more severe cell death occurred in the *fls2* mutant and *rbohd rbohf* double mutant.

### 2.3. FLS2 and RBOHD Are Involved in Regulating Proline Accumulation under Drought Stress

Proline content increases in plants under stress conditions [57,58]. As a ROS scavenger and molecular chaperone, proline is essential for regulating cell osmotic pressure and stabilizing protein structure to protect cells from stress-induced damage [59]. An analysis of the proline content in the leaves of WT and mutants indicated that proline accumulation was significantly increased after drought and salt treatments (Appendix A). Consistent with the growth phenotype under drought stress, the proline content in the *fls2* mutant was significantly lower than that in WT, whereas it was significantly higher in the *rbohd rbohf* double mutant than in WT. However, under salt stress, the proline content in both *fls2* mutant and *rbohd rbohf* double mutant was significantly higher than that of WT.

### 2.4. FLS2 and RBOHD Are Involved in Regulating Transcriptome Reprogramming in Response to Drought and Salt Stress

We next sought to explore the signaling pathways mediated by FLS2 and RBOHD under abiotic stress conditions. An RNA-sequencing analysis was performed using the leaves of WT, *fls2* mutant, and *rbohd rbohf* double mutant treated with drought and salt. We screened DEGs for 12 comparison groups (Appendix A). The largest numbers of DEGs were identified in comparison groups of rbohd/f_NaCl versus rbohd/f_CK (control), rbohd/f_CK versus WT_CK, and rbohd/f_NaCl versus WT_NaCl (Appendix A), suggesting that RBOHD-mediated signals are essential for regulating the transcriptome reprogramming in response to NaCl stress. Consistent with the growth phenotype, the minimum number of DEGs was identified in fls2_drought versus fls2_CK, suggesting that the changes in growth phenotype reflected transcriptome reprogramming. We further performed cluster analysis on DEGs and drew a cluster heatmap and Venn diagram (Figure 2). Results from the heatmaps showed that, compared with WT, the gene expression of *fls2* mutant and *rbohd rbohf* double mutant showed significant changes, especially in *rbohd rbohf* double mutant under control, drought, and salt treatment conditions (Figure 2A–C). These results suggest that FLS2 and RBOHD play important roles in regulating the plant response to drought and salt stress. Further, the results of the Venn diagram revealed certain co-regulatory genes between the DEGs of the mutant and WT under drought and salt conditions, compared with the control (Figure 2D,E).

Further, principal component analysis (PCA) showed that the gene expression in three biological repeated samples of WT, *fls2* mutant, and *rbohd rbohf* double mutant clustered closely under normal, drought, and salt stress conditions, showing high biological repeatability (Appendix A). However, the gene expression in WT, *fls2* mutant, and *rbohd rbohf* double mutant was distributed far away from each other on the PCA map, especially the *rbohd rbohf* double mutant that maintained a long distance from WT and *fls2* mutant, indicating significant differences in transcriptome patterns between mutants and WT. Correlation analysis of gene expression among different samples further verified the results of the PCA analysis (Appendix A).

### 2.5. FLS2 and RBOHD Are Involved in the Regulation of Gene Expression Related to Drought and NaCl Stress

To investigate the biological function of DEGs in different comparison groups, we performed gene ontology (GO) analysis on up-regulated and down-regulated DEGs (Figure 3). Among the up-regulated DEGs (Figure 3A and Appendix A), WT_drought versus WT_CK and fls2_CK versus WT_CK shared similar GO terms and clustered in one branch, suggesting that under normal conditions, the effects of the *fls2* mutant on the transcriptome were similar to those of drought treatment on the transcriptome of WT. In another branch, GO terms enriched in WT_NaCl versus WT_CK and rbohd _CK versus WT_CK clustered in one branch (Figure 3B and Appendix A), indicating that under normal growth conditions, the expression of genes related to salt stress was activated in the *rbohd rbohf* double mutant. Heatmap analysis of DEGs showed that compared with WT_CK, certain representative genes expressed in WT_ Drought and fls2_CK showed a similar trend (Figure 3C). Under normal growth conditions, the expression of certain representative genes in *rbohd/f*_CK was similar to that in WT_NaCl (Figure 3D). Compared with WT_CK, the expression of genes related to JA increased after salt treatment; moreover, the expression of genes related to JA in *rbohd/f_*CK was activated, indicating that under normal growth conditions, the *rbohd rbohf* double mutant showed a stress response similar to that under NaCl treatment due to defective H_2_O_2_ signaling (Figure 3A,D). The expression of genes related to cellular response to hypoxia, bacterium, innate immune response, and oxidative stress was significantly up-regulated in *rbohd/f_*NaCl, compared with that in *rbohd/f_*CK (Figure 3A). Interestingly, the expression of genes related to response to oxidative stress in *rbohd/f_*CK was activated compared with that of WT_CK, indicating that defective H_2_O_2_ production in the *rbohd rbohf* double mutant changed the intracellular ROS homeostasis, consequently resulting in an oxidative stress response (Figure 3A). In the GO analysis of down-regulated DEGs (Figure 3B), under drought treatment, the expression of genes related to the cellular response to hypoxia was significantly inhibited in the *rbohd rbohf* double mutant, indicating that the *rbohd rbohf* double mutant could not respond to intracellular hypoxia under drought stress. These results showed that genes induced and expressed in plants under abiotic stress are widely involved in biotic stress response.

### 2.6. FLS2 and RBOHD Are Involved in Regulating Plant Response to Drought and Salt Stress

Compared with the control, the growth of WT and *rbohd rbohf* double mutant changed after 1 week of drought and salt stress treatments, especially the growth of *rbohd rbohf* double mutants was significantly inhibited (Figure 1). These changes in growth phenotype reflect the compromise made by plants in response to environmental stresses. Changes in plant growth can lead to reprogramming of the transcriptome to adapt to a new growth environment. To analyze the effects of drought and salt stress on the transcriptome, we compared and analyzed the DEGs in WT, *fls2* mutant, and *rbohd rbohf* double mutant under control, drought, and salt treatment conditions, respectively. The results of heatmap analysis of DEGs showed that, compared with WT, the expression of genes in response to salt stress and water deprivation was significantly increased in *rbohd rbohf* double mutant under normal growth conditions with no significant change in the *fls2* mutant (Figure 4). This is consistent with the result showing that the growth state of *fls2* mutant was similar to that of WT under normal growth conditions (Figure 1). Under normal growth conditions, genes related to salt response and water privatization were highly expressed in *rbohd rbohf* double mutant (Figure 4), indicating that RBOHD-mediated H_2_O_2_ signaling plays a key role in regulating the stress response of plants to salt and drought. Under salt stress, the expression of genes related to salt stress response and protein kinase activity in *rbohd rbohf* double mutant was significantly higher than that in WT. In contrast, the expression of genes involved in salt stress response in *fls2* mutant was lower than that in WT (Appendix A). Consistent with the growth phenotype, under drought treatment, the expression of genes related to drought stress response in *rbohd rbohf* double mutant was significantly higher than that in WT, whereas the expression of certain genes related to drought stress response in *fls2* mutant was lower than that in WT (Figure 5A). The GO enrichment analysis showed that under drought treatment, DEGs in the *rbohd rbohf* double mutant were primarily involved in regulating plant water deficit response compared with WT (Figure 5B).

### 2.7. PIF4 might Act Downstream of FLS2 and RBOHD Signaling 

To explore the regulatory mechanism of FLS2- and RBOHD-mediated signals on the expression of downstream genes, we screened transcription factors (TFs) from DEGs in different comparison groups (Figure 6) and constructed a regulatory network based on these TFs (Figure 6). Interestingly, we found that PIF4 was the primary TF in WT_Drought versus WT_CK, and its expression was regulated by several TFs (Figure 6A). In order to examine whether the expression of *PIF4* can be induced by drought or NaCl treatment, we obtained the transgenic plants expressing the *PIF4pro::GUS.* The seedlings of *PIF4pro::GUS* were grown on 1/2 MS medium plates containing 150 mM mannitol and 100 mM NaCl for 7 days, and untreated seedlings were used as controls. Then the GUS activities were detected. The results showed that the expression of *PIF4pro::GUS* was significantly induced under mannitol and NaCl stress, compared with that of control (Figure 7A). In addition, compared with WT, the expression of *PIF4* was induced in *fls2* mutant and *rbohd rbohf* double mutant only under drought treatment; however, the increased proportion was significantly lower than that in WT (Figure 7B). Under NaCl treatment, the expression of *PIF4* in *fls2* mutant and *rbohd rbohf* double mutant decreased (Figure 7B). As expected, the seedlings of *pif4* showed increased tolerance to both drought and salt stress (Figure 7C). To investigate whether PIF4 responded to FLS2-dependent signaling, we examined the sensitivity of *pif4* mutant to flg22. As shown in Appendix A, the flg22 treatment strongly inhibited the root elongation of WT seedlings but only slightly suppressed the root elongation of *pif4* seedlings. These results suggested that PIF4 may act downstream of FLS2-dependent signaling to mediate the immune response. Altogether, these results suggest that FLS2- and RBOHD-mediated signals are involved in regulating *PIF4* expression under drought and NaCl treatments. Further, the analysis showed that *pif4* mutant showed tolerance to drought and salt stress (Figure 1). These results suggested that FLS2- and RBOHD-mediated signals regulate the response of plants to drought and salt stress by regulating the expression of *PIF4*.

## 3. Discussion

### 3.1. FLS2 Is Involved in Regulating Plant Response to Drought Stress

Compared with WT, *fls2* mutant showed decreased sensitive to drought and salt stresses (Figure 1), which was very similar to the insensitive of *fls2* mutant to immune stress [36,37]. FLS2 is an important receptor kinase of flg22, a conserved 22-amino-acid peptide of plant–pathogen [36,37]. In recent years, certain important signal peptides and receptors for regulating plant response to drought were discovered [26,29]. The decreased sensitive of *fls2* mutant to drought reflected the inability of *fls2* mutant to perceive drought and salt stress. Considering the function of FLS2 as a receptor kinase, based on our results here, we proposed that FLS2 is involved in the perception of drought stress (Figure 1, Figure 3, and Figure 4). Further experimental verification of this potential function of FLS2 will provide new insights into the mechanism by which plants deal with compound stress.

### 3.2. The rbohd rbohf Double Mutant Showed Hypersensitive phenotype to Drought and Salt Stress

Unlike *fls2* mutant, *rbohd rbohf* double mutant showed a hypersensitive phenotype to drought and salt stress (Figure 1). The RNA-seq analysis further showed that under drought and salt stress, the expression of genes related to abiotic stress in *rbohd rbohf* double mutant was significantly higher than that in WT (Figure 3, Figure 4 and Figure 5). In addition, the growth of *rbohd rbohf* double mutant in soil was significantly slower than that of WT under normal growth conditions (Figure 1B). The genes related to drought and salt stresses were abnormally activated in *rbohd rbohf* double mutant even under normal growth conditions in soil (Figure 3, Figure 4 and Figure 5). DAB staining analysis showed that *rbohd rbohf* double mutant was impaired in accumulating H_2_O_2_ under both normal and stress conditions in soil (Appendix A). ROS plays a key role in regulating plant response to drought and salt stress [29]; these results show that plants may still endure a certain degree of drought and salt stress when grow in soil even under normal growth conditions. An analysis of the transcriptomes of WT, *fls2* mutant, and *rbohd rbohf* double mutant under different growth conditions preliminarily revealed the potential mechanism by which FLS2 and RBOHD regulate the response of plants to drought and salt stresses.

### 3.3. FLS2–PIF4 Module Is Involved in Regulating the Response of Plants to Compound Stress

Plants exploit the sophisticated signaling pathways to respond to adverse factors, including biological and abiotic stresses [15]. Although different signaling pathways regulate the corresponding biotic and abiotic stresses, these two kinds of stresses often occur simultaneously, thereby aggravating their effect on plants. For example, under drought stress, the occurrence of diseases and pest infestation increase [7,12,19,20,60]. This activates multiple or single pathways in plants to cope with these stresses [19,20,24]. Both biotic and abiotic stresses can lead to the accumulation of ROS and fluctuation in Ca^2+^ levels in cells [32,35,61], suggesting that ROS or calcium signal is required to regulate the response of plants to compound stress. In response to pathogenic microbes, FLS2 and RBOHD are key regulatory factors regulating cellular ROS, calcium signals, and H_2_O_2_ produced by RBOHD, which plays a crucial role in regulating abiotic stress [46,48,51]. However, studies on the effects of ROS and calcium signals regulated by FLS2 and RBOHD on the transcriptome under abiotic stress are highly limited. An analysis of the responses of *fls2* mutant and *rbohd rbohf* double mutant to drought and salt stress revealed that *fls2* mutant exhibited tolerance to abiotic stress similar to immune stress (Figure 1B). To investigate the potential roles of FLS2 and RBOHD in the regulation of transcriptome during the response to abiotic stress, we performed RNA-seq analysis on the drought and NaCl treated seedlings of fls2, *rbohd rbohf* double mutant and WT (Figure 2). The RNA-seq analysis showed that FLS2- and RBOHD-mediated signals were widely involved in regulating gene expression related to drought and salt stresses (Figure 3, Figure 4 and Figure 5).

Further TF network analysis showed that FLS2 and RBOHD regulate the response of plants to drought and salt stress by regulating the expression of *PIF4* (Figure 6). In addition, our earlier studies confirmed that PIF4 and WRKY33 regulate the homeostasis of cellular ROS of plants under drought and salt stress [62]. WRKY33 is an important transcription factor downstream of FLS2- and RBOHD-mediated signals, which is involved in mediating the immune response of plants to pathogens [63,64,65]. Therefore, our results revealed that the signaling pathway FLS2–RBOHD–PIF4 regulated plant response to biotic and abiotic stresses.

Taken together, in this study, we revealed the potential roles of FLS2 and RBOHD in the regulation of transcriptome in response to the abiotic stresses, although the functions of these two proteins in the regulation of immune response and ROS production have been extensively characterized. We identified the DEGs and the TF networks that are involved in regulating the abiotic stress response of higher plants. Based on the TF network, we identified one important TF, PIF4, which showed potential roles in the regulation of abiotic stress response.

## 4. Materials and Methods

### 4.1. Plant Material and Growth Conditions

THE *fls2* (SALK_099606 and SALK_141277), *pif4* (SAIL_1288_E07), and *rbohd rbohf* double mutants (CS9558 and CS68522) were obtained from the *Arabidopsis* Biological Resource Center (Appendix A). Homozygous T-DNA insertion lines were confirmed by polymerase chain reaction (PCR) using gene-specific and T-DNA-specific primers (Appendix A). For flg22 treatment, 5-day-old seedlings were transferred into 1/2 Murashige and Skoog (MS) liquid medium plus 1 μM flg22 (a synthetic peptide of 22 amino acids) (PhytoTech, Lenexa, KA, USA) for further growth for 1 week. Liquid 1/2 MS medium without flg22 was used as a mock. All mutant and WT plants were grown in climate chambers at 22 °C and 100 µM photons m^−2^ s^−1^ in a 16 h light/8 h dark cycle [66,67]. For NaCl treatments, 1-week-old seedlings were first transplanted in the soil to grow for 1 week under normal growth conditions. Subsequently, these were watered with an aqueous solution containing 100 mM NaCl and allowed to grow further for 1 week. For drought treatment, the seedlings were first transplanted into normal watered soil; afterward, watering was stopped after transplantation. After 1 week, the soil water content decreased to about 10% and the seedlings were further grown for 1 week. For control, seedlings of the same batch were transplanted into the soil and grown under normal watering conditions (watering once a week) for 3 weeks.

### 4.2. Detection of Reactive Oxygen Species

The levels of O_2_^−^ and H_2_O_2_ were detected by Nitro Blue Tetrazolium (NBT) and 3,3′ Diaminobenzidine Tetrahydrochloride (DAB) staining, as described by Guo et al. [68]. Seedlings were vacuum infiltrated with either 0.5 mg mL^−1^ NBT in 10 mM potassium phosphate buffer (pH 7.8) or 1 mg mL^−1^ DAB in distilled H_2_O adjusted to pH 3.8 by adding HCl. The seedlings were then incubated in the dark at room temperature for 1 h (NBT) or under light for 8 h at 30 °C (DAB). After staining, the seedlings were incubated in 100% ethanol to remove chlorophyll.

### 4.3. RNA-Seq Analysis

Total RNA was extracted using the mirVana miRNA Isolation Kit (Ambion) following the manufacturer’s protocol. RNA integrity was evaluated using an Agilent 2100 Bioanalyzer (Agilent Technologies, Santa Clara, CA, USA). The samples with RNA Integrity Number (RIN) ≥ 7 were subjected to subsequent RNA-seq analysis. The libraries were constructed using TruSeq Stranded mRNA LTSample Prep Kit (Illumina, San Diego, CA, USA) according to the manufacturer’s instructions. Libraries were sequenced on the Illumina sequencing platform (HiSeqTM 2500 or Illumina HiSeq X Ten), and 125-bp/150-bp paired-end reads were generated. Then, Raw data (raw reads) were processed using Trimmomatic. The reads containing ploy-N and the low quality reads were removed to obtain the clean reads. Then the clean reads were mapped to the reference genome using hisat2. After that, the FPKM value of each gene was calculated using cufflinks, and the read counts of each gene were obtained by htseq-count. DEGs were identified using the DESeq (2012) R package functions estimateSizeFactors and nbinomTest. *p* value < 0.05 and fold Change > 2 or fold Change < 0.5 was set as the threshold for significantly differential expression. Hierarchical cluster analysis of DEGs was performed to explore genes expression patterns. RNA sequence data are available at https://dataview.ncbi.nlm.nih.gov/?search=SUB8234436; accessed on 1 January 2022.

### 4.4. GO and KEGG Analysis

The KEGG and GO analyses were performed on Metascape: http://metascape.org; accessed on 1 January 2022. Firstly, the DEGs in each comparison group were screened. Then the DEGs list was uploaded on Metascape according to the operating manual. After uploading the data, both GO and KEGG analyses were performed on Metescape automatically. After the analysis was completed, the results of GO and KEGG were downloaded from the Metascape.

### 4.5. Construction of the Transcription Factor Network

The transcription factors were identified from the DEGs of each comparison group according to the Arabidopsis transcription factor database, which can be downloaded from PlantTFDB (http://planttfdb.cbi.pku.edu.cn/; accessed on 1 January 2022). Then, the regulatory relationship between a transcription factor and target genes was determined according to the interaction database, which can be downloaded from http://plantregmap.gao-lab.org/download.php; accessed on 1 January 2022. Then, the transcription factor network was constructed using the cytoscape program (http://www.cytoscape.org/; accessed on 1 January 2022).

### 4.6. GUS Staining and Histological Analysis

Histochemical GUS staining was performed with GUS staining Kit, according to the manual (G3061, Solarbio Co., Beijing, China). Samples were fixed in 90% acetone at −20 °C, rinsed four times with 0.1 M sodium phosphate buffer (pH 7.4), and then incubated in X-Gluc solution [0.1 M sodium phosphate (pH 7.4), 3 mM potassium ferricyanide, 0.5 mM potassium ferrocyanide, 0.5 g L^−1^ 5-bromo-4-chloro-3-indolyl-β-d-glucuronide cyclohexilammonium salt] at 37 °C. After staining, samples were incubated in methanol to remove chlorophyll and then mounted in the clearing solution (a mixture of chloral hydrate, water, and glycerol in a ratio of 8:2:1). Observation was performed using a stereomicroscope (MZ16F, Leica Microsystems, Germany) or a microscope equipped with Nomarski optics (BX51, Olympus Co., Tokyo, Japan).

## Figures and Tables

**Figure 1 ijms-23-01080-f001:**
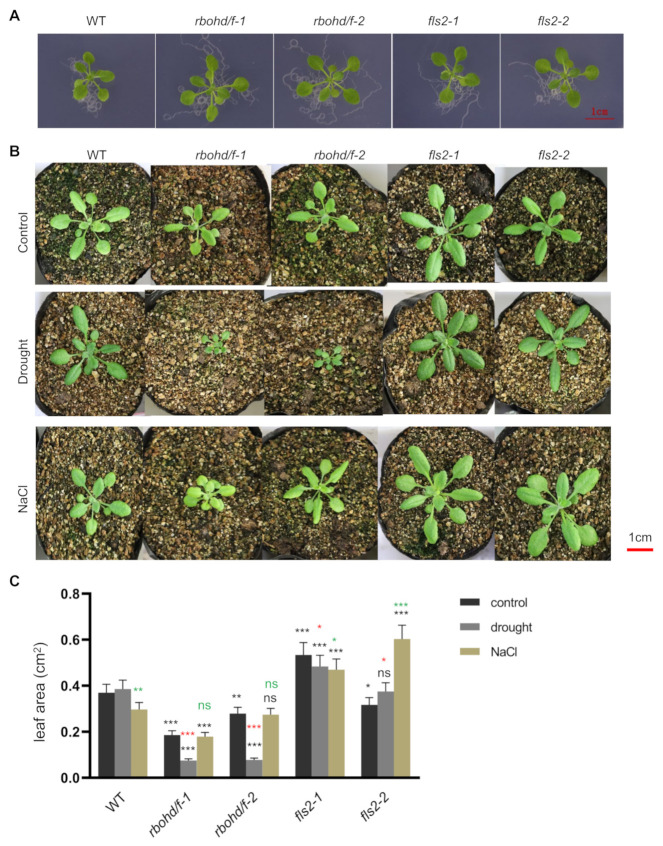
The sensitivity analysis of *fls2* mutant and *rbohd rbohf* double mutant to drought and salt stress. (**A**) The one-week-old seedlings of WT, *fls2* and *rbohd rbohf* double mutants (*rbohd/f*) were grown on 1/2 MS medium plates under normal conditions. (**B**) Two-week-old seedlings of WT and mutants were treated with drought and NaCl for one week, and untreated seedlings were used as controls. Scale bar: 1 cm. (**C**) Statistical analysis of the leaf area of seedlings of WT and mutants in (**B**). The data were analyzed by one-way ANOVA following Brown–Forsythe test. ns: *p* > 0.05, *: *p* < 0.05,**: *p* < 0.01, ***: *p* < 0.001. The black stars represent the comparison between mutant and WT; red stars represent the comparison between drought and control; green stars represent the comparison between NaCl and control.

**Figure 2 ijms-23-01080-f002:**
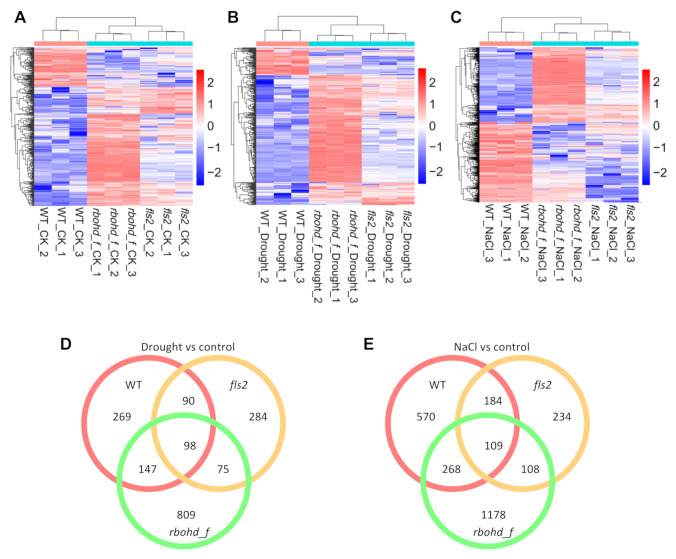
Heatmap and Venn analysis of DEGs under drought and salt conditions. (**A**–**C**) The heatmap analysis of the expression of DEGs. The number of suffix of sample name represents 1–3 repetitions. The screening criteria were fold change > 2 and *p*-value < 0.05. Red indicates gene up-regulation, blue indicates gene downregulation, and color depth indicates the degree of difference. (**D**,**E**) The Venn analysis compares the co-regulatory number of DEGs under drought and salt treatment with that of control.

**Figure 3 ijms-23-01080-f003:**
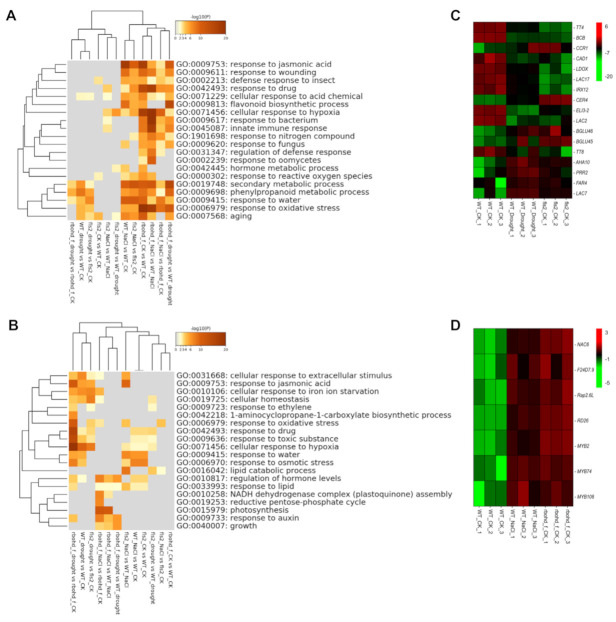
The GO enrichment analysis of differentially expressed genes in different comparison groups. (**A,B**) The GO enrichment analysis of differentially expressed genes (DEGs) in different comparison groups. The up-regulated and down-regulated DEGs were analyzed by GO. The closer the biological functions of DEGs are, the more they will cluster into the same branch. (**C**,**D**) The heatmap analysis of representative DEGs related to response to water deprivation and salt stress. Red indicates the up-regulation of gene expression, green indicates the downregulation of gene expression, and the color depth indicates the degree of difference.

**Figure 4 ijms-23-01080-f004:**
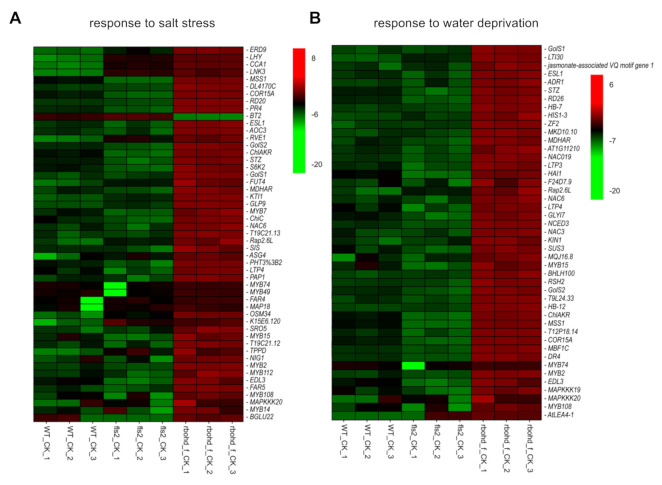
The expression of genes related to drought and salt stress was constitutively activated in *rbohd rbohf* double mutant. (**A**,**B**) Heatmap analysis of genes expression related to salt and water privatization response in WT, *fls2* mutant, and *rbohd rbohf* double mutant under normal growth conditions, respectively. CK: control.

**Figure 5 ijms-23-01080-f005:**
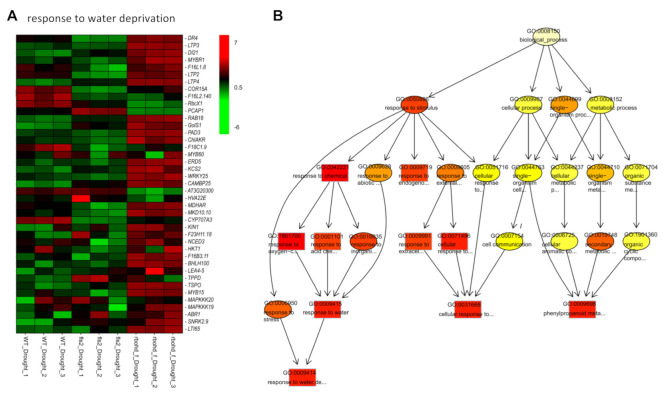
FLS2 and RBOHD are involved in regulating expression of genes related to water shortage under drought conditions. (**A**) Heatmap of gene expression related to water deprivation response in WT, *fls2* mutant, and *rbohd rbohf* double mutant under drought treatment. (**B**) GO enrichment analysis of differentially expressed genes in *rbohd rbohf* double mutants under drought treatment.

**Figure 6 ijms-23-01080-f006:**
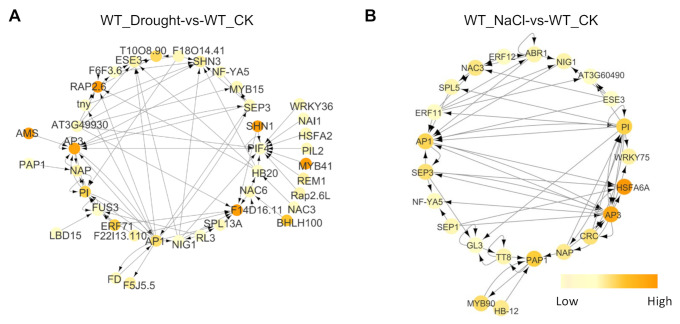
Analysis of transcription factor regulatory network of DEGs in WT under drought and salt stress. (**A**,**B**) The differentially expressed transcription factors in different comparison groups were screened and used to construct their direct mutual regulation network.

**Figure 7 ijms-23-01080-f007:**
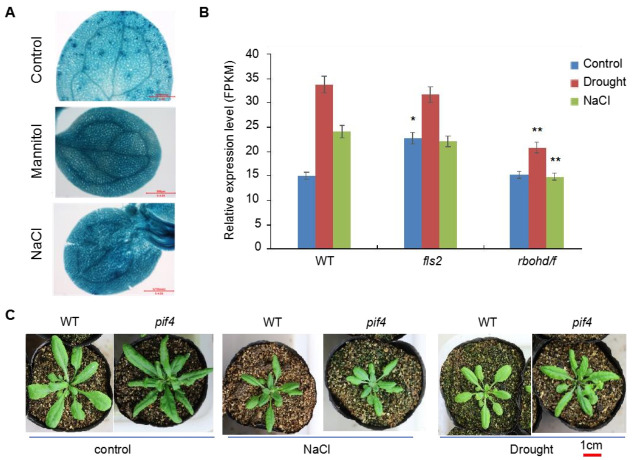
Analysis of the expression of *PIF4* under drought and NaCl conditions. (**A**) Analysis of the GUS activity in cotyledons of *PIF4pro:: GUS* seedlings treated with 300 µM mannitol and 100 mM NaCl. Untreated seedlings were used as controls. The scale bar: 500 μm. (**B**) Statistical analysis of the relative expression of *PIF4* in leaves of WT, *fls2*, and *rbohd rbohf* double mutants under drought and NaCl stress conditions, untreated samples were used as controls. *: *p* < 0.05, **: *p* < 0.01, Student’s *t*-test versus WT. (**C**) Growth of *pif4* and WT under control, NaCl and drought conditions.

## Data Availability

All data supporting the findings of this study are available within the paper and within its Appendix A published online.

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
