# Peer review of "FLS2–RBOHD–PIF4 Module Regulates Plant Response to Drought and Salt Stress"

_ijms, 2022, doi:10.3390/ijms23031080_

Round 1
Reviewer 1 Report
Its very highly appreciated study to publish, but i have still concerns like,
in figure 7, why in sudden mannitol treatment has been come where overall manuscript was about direct drought by stopping watering.
another one is that at line number 366-368 under section 4, why watering has been stop after transplanting? and what was the actual duration of this treatment. most importantly if you have continued with such 10% of water content for next 1 week, then what was the status of water content. author need to write details in this section.
Conclusion has to be included.
Reviewer 2 Report
The MS focused on molecular mechanism of salt and drought stress in plants using respiratory burst oxidase homolog (RBOHD) and flagellin sensitive 2 (FLS2) mutants. Both ROS and calcium signaling is required for early sensing and responses to abiotic stresses. From the mutant analysis, it is clear that mutation in RBOHD/F impaired the ROS generation which is important for stress sensing and response in plants. The authors nicely discussed growth, and associated biochemical analysis in these mutants, which was further complemented with transcriptome analysis. However, I have following concerns with the MS and needs rectification.
1. Introduction Line 54 to Line 70 needs rewriting to clarify how FLS2 and RBOH interact to produce ROS. Whether their (FLS2 and RBOH) interaction in ROS generation is necessary.
In addition, spell out abbreviated terms first. BAK1, BIK1 ... PUB12, PUB13 ... Not Clear
Line 72 ... What is meant by immune stresses .... Do you mean biotic stresses or Immune responses ... not clear
Line 73-77 .... Objectives of the study are not clear. What is the question or hypothesis?
Discussion: First two parts are some what confusing due to the words "sensitivity" and "sensing" Mutants become hypersensitive --- mean highly stress sensitive or the authors means they sense the stress at a greater rate.
Please do clarify it. Moreover, Discussion part is smaller as most of the results were partially discussed in RESULTS section. It is suggested to write the results in RESULTS section and while Discussion part should be extensive, interactive.
Methods: Transcriptome analysis part is incomplete. How RNA sequence quality was checked? How the reads were cleaned or rome the adapter sequences? How the reads were aligned? Which reference genome was used? Which software/platform Windows/Linux was used? What about Feature Counts and DEGs? What about KEGG analysis, GO analysis? How the heat maps were generated? How the cluster analysis with heat map were carried out? How TF networking analysis was done? Any Pathway analysis? Which software was used?
Overall the MS can be accepted after suggested revisions.
Round 2
Reviewer 1 Report
Accepted